# Public attitudes toward allocating scarce resources in the COVID-19 pandemic

Wesley Buckwalter[1], Andrew Peterson[2]*

**1** Department of Philosophy, University of Manchester, Manchester, United Kingdom, **2** Department of Philosophy, Institute for Philosophy and Public Policy, George Mason University, Fairfax, Virginia, United States of America

* apeter31@gmu.edu

**Data Availability Statement:** All data, and detailed materials and coding, are available in the Supporting information and through an Open Science Foundation project (https://osf.io/cdtm6/).

## Abstract

The general public is subject to triage policies that allocate scarce lifesaving resources during the COVID-19 pandemic, one of the worst public health emergencies in the past 100 years. However, public attitudes toward ethical principles underlying triage policies used during this pandemic are not well understood. Three experiments (preregistered; online samples; N = 1,868; U.S. residents) assessed attitudes toward ethical principles underlying triage policies. The experiments evaluated assessments of utilitarian, egalitarian, prioritizing the worst-off, and social usefulness principles in conditions arising during the COVID-19 pandemic, involving resource scarcity, resource reallocation, and bias in resource allocation toward at-risk groups, such as the elderly or people of color. We found that participants agreed with allocation motivated by utilitarian principles and prioritizing the worst-off during initial distribution of resources and disagreed with allocation motivated by egalitarian and social usefulness principles. At reallocation, participants agreed with giving priority to those patients who received the resources first. Lastly, support for utilitarian allocation varied when saving the greatest number of lives resulted in disadvantage for at-risk or historically marginalized groups. Specifically, participants expressed higher levels of agreement with policies that shifted away from maximizing benefits to one that assigned the same priority to members of different groups if this mitigated disadvantage for people of color. Understanding these attitudes can contribute to developing triage policies, increase trust in health systems, and assist physicians in achieving their goals of patient care during the COVID-19 pandemic.

## Introduction

Public health emergencies can place a significant burden on health systems [1–4]. Patients needing lifesaving treatment may quickly outpace a hospital's resource capacity. When this happens, decisions about how to allocate scarce resources must be made. Triage policies provide guidance in these circumstances [5]. They outline specific rules for distributing scarce resources, such as ventilators, ICU beds, hospital staff, and other lifesaving resources among

**Funding:** AP is supported by the Greenwall
Foundation Faculty Scholars Program.

**Competing interests:** The authors have declared
that no competing interests exist.

patients in need. Similar reasoning may also guide the distribution of novel therapeutics or
vaccines, when they become available.

Triage policies are motivated by different, and sometimes competing, ethical principles.
Most appeal to the utilitarian principle of maximizing benefits [6–8]. Triage policies that aim
to maximize benefits involve allocating scarce resources to patients to save the greatest number
of lives or preserve the largest amount of life-years among treated patients. Triage policies
might also be motivated by different ethical principles, such as egalitarianism, prioritizing the
worst-off, or social usefulness [9]. Egalitarianism gives all patients an equal chance at receiving
scarce resources. Triage policies motived by egalitarianism might allocate resources through a
random lottery or a first-come-first-served approach. Prioritizing the worst-off allocates scarce
resources to patients in order of the seriousness of their condition or susceptibility to disease.
Lastly, the principle of social usefulness prioritizes patients who have the greatest prospective
value to society. A triage policy motivated by this principle might prioritize front-line health
care workers or research participants in vaccine trials. While there is scholarly debate about
which ethical categories best reflect different allocation principles (see General discussion), we
retain these terms and descriptions to begin distinguishing between principles in ordinary
judgment.

Principles of resource allocation remain matters of practical and public debate [9–13]. Triage frameworks are motivated by multiple allocation principles, but the principle of maximizing benefits is often the central approach around which other allocation principles are
organized. However, as many have argued, this approach can also reveal or even perpetuate
latent attitudes and structural inequalities that favor the allocation of scarce resources to some
patients over others [14]. If certain patient groups have a lower chance of surviving treatment,
due to pre-existing comorbidities, biological age, disability, or structural racism in the social
determinants of health, they may receive less priority than others for lifesaving resources [10,
11, 15–17]. This raises concerns about the fairness of resource allocation in health emergencies, and public trust in health care systems.

Concerns about fairness and trust highlight the need for public involvement in developing
and evaluating triage policy. Triage policies can be enhanced by consulting the communities
impacted by them [18]. Failure to address public concern through "closed door" policy development could sow distrust and negatively affect patient care. This worry is pronounced in the
COVID-19 pandemic, as Black Americans are dying at significantly greater rates than White
Americans [17] and mortality in long-term care facilities is also disproportionately high [19,
20].

Prior to COVID-19, some research groups studied public reaction to triage policies through
survey studies and community engagement [21–25]. While these efforts provide valuable
insight into public opinion, further study in the current pandemic could provide vital information for health systems. First, there is a palpable worry that, as compliance with physical distancing and mask recommendations wanes, and the flu season approaches, COVID-19
caseloads might increase in hotspots across the United States and lifesaving resources may
again become scarce. Second, there has been international discussion on the ethics of withdrawing ventilators or other scarce resources from patients with poor prognoses to reallocate
them to others with better chances of survival [26], but this practice remains controversial.
Third, there has been public outcry over racial disparities in COVID-19 related infections and
deaths in the United States and abroad, highlighting the need for potential changes to health
care policy that promote justice [14]. Lastly, frameworks for the allocation of other scarce
resources, such as vaccines, will soon be the focus of debate in the United States and internationally. Public attitudes toward ethical principles underlying resource allocation in these circumstances are not well understood.

The present study seeks to understand how laypeople evaluate allocation principles, with focus on triage dilemmas posed by the COVID-19 pandemic. Our specific research questions are as follows. First, what effect does scarcity during the pandemic have on lay evaluations of utilitarian, prioritizing the worst-off, egalitarian, or social usefulness principles (Experiment 1)? Second, does the same pattern of judgments made about allocation principles involving new patients extend to reallocating resources between existing patients (Experiment 2)? And third, does the public support a utilitarian allocation principle even if it disadvantages members of at-risk or historically marginalized groups, such as people with disabilities or people of color (Experiment 3)?

## General methods

We conducted three preregistered experiments, involving a total of 1,868 U.S. residents, between April and May 2020, when the COVID-19 death toll in the United States exceeded 100,000. All participants were adult residents of the United States. Participants were recruited and tested using the online platforms, Mechanical Turk (https://www.mturk.com) and Qualtrics (https://www.qualtrics.com). All experiments were approved by the University of Manchester University Research Ethics Committee (UREC #9628).

All data and materials for each experiment are available in the supporting information (S1 and S2 Files) and through the Open Science Framework (https://osf.io/cdtm6/). No previous literature provided guidance on target recruitment for power analysis. Consistent with past survey studies, target recruitment was set at approximately 75 participants per condition within the experiments (or 600 per experiment), with sufficient additional participants as a precaution against attrition.

Each experiment was designed to address one of the three above research questions. Experiments contained scenarios and manipulations derived from literature review and expert knowledge of triage dilemmas posed by the COVID-19 pandemic. These dilemmas involve allocation in conditions of scarcity, the forcible withdrawal of resources from patients with poor prognoses to give to others with better prognoses, and the treatment of populations who may be disadvantaged in receiving scarce resources. Consistent with our preregistration, participant attitudes were evaluated using a combined measure of agreement with allocation principle and the behavioral intentions regarding which hospital participants would choose to receive lifesaving treatment.

## Experiment 1

Experiment 1 tested the effect of resource scarcity on layperson attitudes toward principles of resource allocation.

### Methods

Six hundred and twenty-two people participated in the study. Their mean age was 37 years (range = 18–77, SD = 12), 36% were female, and 95% reported native competence in English (see S1 for complete demographic details). Participants were randomly assigned to one of 8 conditions in a 2 (Resources: plentiful, scarce) × 4 (Principle: order, serious, lives, importance) experimental design. Participants first read a single brief scenario, then responded to two comprehension check questions and two test statements. In each scenario, a hospital adopts a new triage policy for distributing lifesaving resources during a pandemic. The pandemic is described as fast moving and the disease potentially life threatening without adequate medical care. Each triage policy privileges one of the four ethical principles outlined above: prioritizing the worst-off (serious), utilitarian (lives), egalitarianism (order), and social usefulness

**Table 1. Experiment 1: Manipulation of principle.**

| Condition | Principle |
|---|---|
| Serious | According to this policy, patients will receive lifesaving resources in the order of the seriousness of illness, with those who are the worst-off being prioritized. |
| Lives | According to this policy, patients will receive lifesaving resources in the order that saves the most lives, with those having the best chances of recovery being prioritized. |
| Order | According to this policy, patients will receive lifesaving resources in the order in which they arrive, with those arriving to the hospital first being prioritized. |
| Important | According to this policy, patients will receive lifesaving resources in the order of their importance, with those standing to contribute the most to society being prioritized. |

(important). Participants were randomly assigned a condition featuring one of four principles (Table 1). After reading about a principle, participants were further assigned to conditions in which lifesaving resources were either plentiful or scarce. Specifically, they were told that either "there are many more [lifesaving resources available than there are patients/patients than there are lifesaving resources available]" and that because of this "[no/many] patients will go without lifesaving treatment who need them."

Participants were told that a hospital has decided on a new policy for allocating resources, built around one of the four allocation principles. Beneath the scenario, on the same page, participants responded to four statements (order fixed):

1. There are enough lifesaving resources for every patient who needs them. (comprehension 1)

2. Lifesaving resources are allocated to patients based on _________. (comprehension 2)

3. The allocation policy the triage team decided on is a good one. (policy; test 1)

4. I would choose this hospital over others if I or someone I loved needed lifesaving treatment. (hospital; test 2)

Comprehension 1 assessed whether the resource manipulation was effective (i.e. that participants recognized whether resources were plentiful or scarce). Responses were collected by dichotomous yes/no answers randomly rotated. Comprehension 2 assessed whether the principle manipulation was effective (i.e. that participants recognized a specific principle was implemented). Responses were collected by a sentence completion task with the four randomly rotated options, "the order in which they arrive", "the seriousness of illness", "what saves the most lives", or "how important the patient is to society". Participants then responded to two test questions, policy and the hospital, using a standard 7-point Likert scale, 1 ("strongly disagree")—7 ("strongly agree"), arranged left-to-right on the participant's screen. These items evaluated allocation principles on two dimensions: judgments concerning the policy itself and resulting behavioral intentions, such as choosing which hospital to visit in a public health emergency.

## Results

Our research question asked which allocation principles are viewed favorably or unfavorably during resource scarcity. To answer this question, we first excluded those participants who did not pass both comprehension checks (see preregistration). One hundred and eighty-two participants were removed for not answering the comprehension checks correctly, leaving a total sample of four hundred and forty participants. We examined the relationship between responses to our two dependent variables, policy and hospital, using a Spearman-Brown

**Table 2. Experiment 1: Multiple linear regression predicting the combined measure.**

| Predictors | Beta | CI | p |
|---|---|---|---|
| (Intercept) | 5.71 | 5.14 6.29 | < .001 |
| Resources | -0.62 | -1.14–0.11 | .017 |
| Principle | | | |
| Lives | -0.39 | -0.92 0.13 | .139 |
| Order | -0.67 | -1.19–0.16 | .011 |
| Important | -2.38 | -2.91–1.85 | < .001 |
| Sex | 0.21 | -0.05 0.47 | .117 |
| Age | < 0.00 | -0.01 0.01 | .630 |
| Resources*Lives | -0.16 | -0.89 0.57 | .666 |
| Resources*Order | -0.96 | -1.67–0.25 | .008 |
| Resources*Important | -0.51 | -1.23 0.21 | .162 |

N = 440, $R^2$ = 0.4265, Adjusted $R^2$ = 0.4145. Reference class for Resources is Plentiful; reference class for Principle is Serious; reference class for Sex is Female.

prediction formula. The Spearman-Brown test coefficient was .886. As specified in our preregistration, we combined these variables into a single measure for analyzing participant reactions to triage policies along these dimensions. Multiple linear regression was conducted with the combined measure as response and with resources, principle, an interaction of resources and principle, self-reported biological sex, and participant age as predictors.

There were statistically significant effects for resources, principle, and their interaction (Table 2 and Fig 1). Resource conditions were significant predictors of responses, with lower scores in scarce as compared to plentiful conditions (β = -0.62). Principle also predicted responses with lower scores associated with order (β = -0.67) and important (β = -2.38) as compared to serious conditions. The differential impact of resources on the order condition

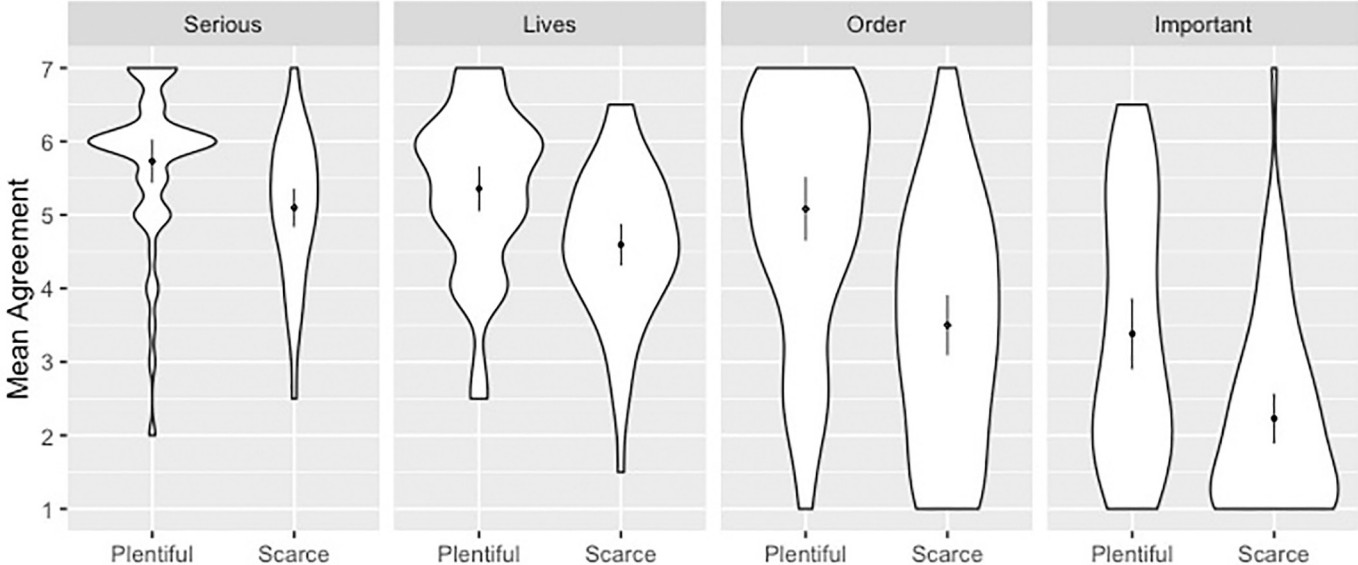

**Fig 1. Mean response for each allocation principle grouped by level of resources.** Scales ran 1 (SD)– 7 (SA). Dots overlay distributions and show means with 2 SEM.

**Table 3. Experiment 1: Independent sample t-tests.**

| | Plentiful | | Scarce | | | | | | | |
|---|---|---|---|---|---|---|---|---|---|---|
| | M | SD | M | SD | t | df | p | MD | 95% CI | d |
| **Serious** | 5.73 | 1.05 | 5.10 | 0.99 | 3.21 | 105 | .002 | 0.63 | 0.24 1.02 | 0.62 |
| **Lives** | 5.36 | 1.10 | 4.59 | 1.03 | 3.65 | 103 | < .001 | 0.76 | 0.35 1.18 | 0.72 |
| **Order** | 5.08 | 1.61 | 3.50 | 1.63 | 5.29 | 116 | < .001 | 1.58 | 0.99 2.17 | 0.97 |
| **Important** | 3.38 | 1.73 | 2.23 | 1.31 | 3.97 | 108 | < .001 | 1.15 | 0.58 1.73 | 0.76 |

was associated with lower scores as compared to the serious condition (β = -0.96). Sex and age were not significant predictors.

Follow-up pairwise comparisons showed significant differences in responses between plentiful and scarce conditions across all allocation principles (Table 3).

Follow-up one sample t-tests showed that when resources were plentiful, responses were significantly above the neutral midpoint of 4 in the order, serious, and lives conditions, and significantly below the midpoint in the important condition (Table 4).

When resources were scarce, responses were significantly above the midpoint in the serious and lives conditions, and significantly below the midpoint in the order and important conditions. Lastly, when resources were scarce, agreement was strongest in the serious condition. Agreement in the serious condition was significantly greater than in the lives (t(108) = 2.61, p = .01, d = 0.50), order (t(118) = 6.41, p < .001, d = 1.17), and important (t(114) = 13.27, p < .001, d = 2.42) conditions.

An exploratory analysis was conducted to determine the influence of demographic and personal factors on responses. We conducted a multiple regression analysis with the combined measure as the dependent variable that included resources, principle, an interaction of resources and principle, race, sex, age, highest level of education completed, religiosity, politics (two measures: social and economic), and hospital experience (two measures: being admitted to a hospital any time in the last five years, and either working or studying as a health care professional) as predictors. Of these factors, there was weak evidence that politics predicted participant responses. On a 1 (extremely conservative)– 7 (extremely liberal) scale, indicating economic liberalism (e.g. "How economically conservative or liberal are you?") was associated with higher scores on the combined measure (β = .11, t (421) = 1.90, p = .058). However, indicating social liberalism (e.g. "How socially conservative or liberal are you?") was significantly associated with lower scores (β = -.15, t (421) = -2.34, p = .02). General level of education predicted responses, with higher levels of education significantly associated with higher scores

**Table 4. Experiment 1: One sample t-tests.**

| | t | df | p | MD | 95% CI | d |
|---|---|---|---|---|---|---|
| **Plentiful** | | | | | | |
| Serious | 11.64 | 49 | < .001 | 1.73 | 1.43 2.03 | 1.65 |
| Lives | 8.86 | 51 | < .001 | 1.36 | 1.05 1.66 | 1.1 |
| Order | 4.98 | 54 | < .001 | 1.08 | 0.65 1.52 | 0.67 |
| Important | -2.55 | 50 | .014 | -0.62 | -1.1 –0.13 | -0.36 |
| **Scarce** | | | | | | |
| Serious | 8.38 | 56 | < .001 | 1.10 | 0.83 1.36 | 1.11 |
| Lives | 4.19 | 52 | < .001 | 0.59 | 0.31 0.88 | 0.58 |
| Order | -2.44 | 62 | .018 | -0.50 | -0.91 –0.09 | -0.31 |
| Important | -10.38 | 58 | < .001 | -1.77 | -2.11 –1.43 | -1.35 |

(β = .12, t (421) = 2.22, p = .03). Lastly, there was weak evidence for an association with participant sex, with higher scores by males (β = .25, t (421) = 1.87, p = .062). No significant effects were observed with other demographic or personal factors. These effects are exploratory, have not been adjusted for multiple comparisons, and are only included here to guide future research.

## Discussion

This experiment examined the effect of resource scarcity on evaluations of allocation principles. We found that when resources were scarce, participants gave lower scores to all principles. However, the degree to which resource scarcity impacted responses depended on the allocation principle. Scarcity had the largest negative effect on responses in the order condition (d = 0.97) and smallest in serious condition (d = 0.62). Participants generally agreed with utilitarian allocation and prioritizing the worst-off and disagreed with egalitarian and social usefulness principles. When resources were scarce, allocation based on seriousness of condition received the highest ratings of agreement as compared to all other principles (d = 0.5 to 2.4).

## Experiment 2

Experiment 2 tested the effect of reallocating lifesaving resources between existing patients on layperson attitudes toward principles of resource allocation.

## Methods

Six hundred and seven people participated in the study. Their mean age was 38 years (range = 18–77, SD = 13), 45% were female, and 93% reported native competence in English (see S1 for complete demographic details). Participants were randomly assigned to one of 8 conditions in a 2 (Stage: allocation, reallocation) × 4 (Principle: serious, lives, order, important) experimental design. Participants read a single scenario and answered questions similar to those used in Experiment 1. In each scenario, a hospital adopts a new triage policy when resources are scarce. Participants were randomly assigned to conditions in which the policy involved allocating scarce resources to new patients or reallocating scarce resources between existing patients (i.e. taking from one to give to another). Participants were then asked to consider one of four allocation principles. The new patient cases were identical to the scarcity conditions used in Experiment 1. The reallocation condition manipulations are shown in Table 5.

Participants were told that a hospital has decided on a new policy for reallocating resources between existing patients, according to one of the four allocation principles. Beneath the

**Table 5. Experiment 2: Manipulation of principles in relocation conditions.**

| Condition | Principle |
|---|---|
| **Serious** | According to this policy, lifesaving resources will be taken from one patient and given to another patient depending on the seriousness of illness, with those who are the worst-off receiving priority. |
| **Lives** | According to this policy, lifesaving resources will be taken from one patient and given to another patient depending on what saves the most lives, with those having the best chances of recovery receiving priority. |
| **Order** | According to this policy, lifesaving resources will never be taken from one patient and given to another patient for any reason aside from the patient recovering or expiring. |
| **Important** | According to this policy, lifesaving resources will be taken from one patient and given to another patient depending on their importance, with those standing to contribute the most to society receiving priority. |

scenario, on the same page, participants responded to similar comprehension and test items used in Experiment 1:

1. There are enough lifesaving resources for every patient who needs them. (comprehension 1)

2. Lifesaving resources are allocated to new patients based on _________. (comprehension 2, allocation conditions only)

3. Lifesaving resources are reallocated between existing patients based on _________. (comprehension 2, reallocation conditions only)

4. The policy the triage team decided on is a good one. (policy; test 1)

5. I would choose this hospital over others if I or someone I loved needed lifesaving treatment. (hospital; test 2)

Response options were the same as those used in Experiment 1, with the exception of the second comprehension check in the reallocation conditions, which were "the patient recovering or expiring", "the seriousness of illness", "what saves the most lives", and "how important the patient is to society".

## Results

Our research question asked which allocation principles are viewed favorably or unfavorably when reallocating resources between existing patients. To answer this question, we first excluded participants who did not pass both comprehension checks (see preregistration). One hundred and twenty-one participants were removed, leaving a total sample of four hundred and eighty-six participants. We examined the relationship between responses to our two dependent variables, policy and hospital, using a Spearman-Brown prediction formula. The Spearman-Brown test coefficient was .856. We averaged these variables into a single combined measure for analysis. Multiple linear regression was conducted with the combined measure as the dependent variable and with allocation stage, principle, an interaction of stage and principle, self-reported biological sex, and age as predictors.

There were significant effects for allocation stage, principle, and their interaction (Fig 2 and Table 6). Stage was a significant predictor of responses, with lower scores for reallocation as compared to allocation ($\beta = -0.61$). Principle also predicted responses with lower scores associated with order ($\beta = -1.14$) and important conditions ($\beta = -2.01$) as compared to the serious condition. The differential impact of stage on the order condition was associated with higher scores as compared to the serious condition ($\beta = 1.60$). There was weak evidence for an effect of sex, whereby mean response was higher for males (M = 3.83, SD = 1.63) than for females (M = 3.70, SD = 1.65), although this did not reach the conventional threshold for statistical significance (p = .056).

Follow-up independent samples t-tests further revealed that the stage/principle interaction occurred because mean agreement was significantly lower during reallocation as compared to initial allocation for the serious and lives conditions, but was significantly higher for the order condition (Table 7).

Follow-up one sample t-test showed that, at the initial allocation stage, responses were significantly above the neutral midpoint of 4 in the serious and lives conditions, and significantly below the midpoint for the order and important conditions (Table 8). At the reallocation stage, however, responses were only significantly above the midpoint in the order condition.

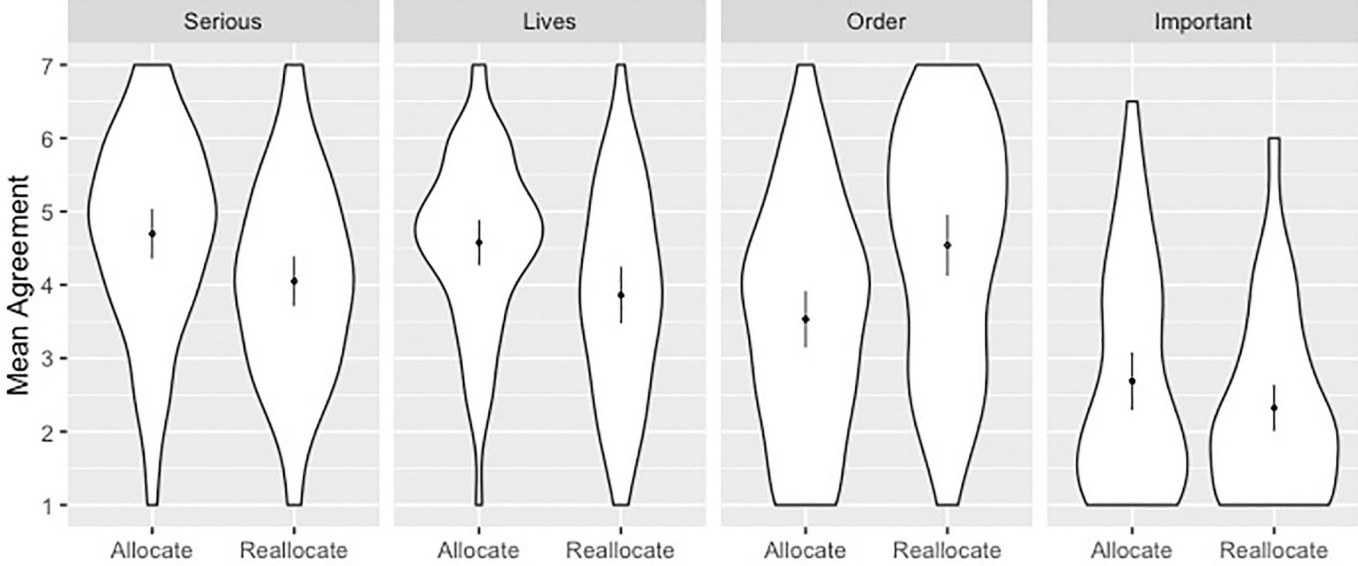

**Fig 2. Mean response for each allocation principle grouped by stage of allocation.** Scales ran 1 (SD)– 7 (SA). Dots overlay distributions and show means with 2 SEM.

**Table 6. Experiment 2: Multiple linear regression predicting the combined measure.**

| Predictors | Beta | CI | p |
|---|---|---|---|
| (Intercept) | 4.63 | 4.06 5.20 | < .001 |
| Stage | -0.61 | -1.11–0.11 | .017 |
| Principle | | | |
| Lives | -0.10 | -0.61 0.41 | .703 |
| Order | -1.14 | -1.64–0.65 | < .001 |
| Important | -2.01 | -2.52–1.50 | < .001 |
| Sex | 0.25 | -0.01 0.51 | .056 |
| Age | < 0.00 | -0.01 0.01 | .666 |
| Stage*Lives | -0.10 | -0.83 0.64 | .793 |
| Stage*Order | 1.60 | 0.89 2.30 | < .001 |
| Stage*Important | 0.23 | -0.48 0.94 | .530 |

N = 484, R2 = 0.269, Adjusted R2 = 0.256. Reference class for Stage is Allocation; reference class for Principle is Serious; reference class for Sex is Female. Two participants were not included in the model because they chose not to disclose their sex or age.

**Table 7. Experiment 2: Independent sample t-tests.**

| | Allocate | | Reallocate | | | | | | | |
|---|---|---|---|---|---|---|---|---|---|---|
| | M | SD | M | SD | t | df | p | MD | 95% CI | d |
| Serious | 4.70 | 1.36 | 4.05 | 1.34 | 2.68 | 124 | .008 | 0.65 | 0.17 1.12 | 0.48 |
| Lives | 4.58 | 1.19 | 3.86 | 1.37 | 2.92 | 107 | .004 | 0.72 | 0.23 1.20 | 0.56 |
| Order | 3.53 | 1.56 | 4.54 | 1.64 | -3.56 | 125 | < .001 | -1.01 | -1.57–0.45 | -0.63 |
| Important | 2.69 | 1.51 | 2.32 | 1.27 | 1.45 | 122 | .149 | 0.36 | -0.13 0.86 | 0.27 |

**Table 8. Experiment 2: One sample t-tests.**

| | t | df | p | MD | 95% CI | d |
|---|---|---|---|---|---|---|
| Allocate | | | | | | |
| Serious | 4.08 | 63 | < .001 | 0.70 | 0.35 1.04 | -0.51 |
| Lives | 3.73 | 58 | < .001 | 0.58 | 0.27 0.89 | -0.49 |
| Order | -2.43 | 64 | .018 | -0.47 | -0.86–0.08 | 0.3 |
| Important | -6.66 | 58 | < .001 | -1.31 | -1.71–0.92 | 0.87 |
| Reallocate | | | | | | |
| Serious | 0.28 | 61 | .777 | 0.05 | -0.29 0.39 | -0.04 |
| Lives | -0.72 | 49 | .473 | -0.14 | -0.53 0.25 | 0.10 |
| Order | 2.59 | 61 | .012 | 0.54 | 0.12 0.96 | -0.33 |
| Important | -10.65 | 64 | < .001 | -1.68 | -1.99–1.36 | 1.32 |

At the reallocation stage, agreement was strongest in the order condition, and this was significantly greater than in the lives (t(110) = 2.35, p = .021, d = 0.45) and important (t(125) = 8.54, p < .001, d = 1.51) conditions. This effect did not pass the conventional threshold for statistical significance in the serious condition (t(122) = 1.83, p = .07, d = 0.33).

An exploratory analysis was conducted to determine the influence of demographic and personal factors on responses. A multiple regression analysis was conducted with the combined measure as the dependent variable that included stage, principle, an interaction of stage and principle, race, sex, age, highest level of education completed, religiosity, social liberalism, economic liberalism, hospital experience, and hospital training as predictors. Political attitudes predicted responses, where social liberalism (e.g. "How socially conservative or liberal are you?") was associated with higher scores on the combined measure (β = -.16, t (466) = -2.55, p = .011). Sex also predicted responses, with higher scores associated with men (β = .26, t (466) = 1.98, p = .048). Hospital training (e.g. "Have you ever worked or studied as a health professional (e.g. nurse, doctor, therapist, etc.)?") was also associated with higher scores (β = .42, t(466) = 2.44, p = .015). No effects were observed with other demographic or personal factors. These effects are exploratory, have not been adjusted for multiple comparisons, and are only included here to guide future research.

## Discussion

This experiment examined the effect that allocation stage (initial allocation versus reallocation) has on attitudes toward allocation principles. We found that the only evaluation positively affected by allocation stage was order. Participants disagreed that resources should be initially allocated to new patients based on order; resources should instead be allocated according to utilitarian principles or prioritizing the worst-off. At the reallocation stage, however, participants were split as to whether resources should be taken from one patient and given to another. At this stage, the approach that received significant agreement was reallocation only after the recovery or death of a patient who initially received resources.

## Experiment 3

Experiment 3 tested the effect of disadvantage for at-risk or historically marginalized patient groups on layperson attitudes toward a utilitarian allocation principle.

## Method

Six hundred and thirty-nine people participated in the study. Their mean age was 39 years (range = 18–89, SD = 13), 38.5% were female, and 94.2% reported native competence in

English (see S1 for complete demographic details). Participants were randomly assigned to one of 8 conditions in a 2 (Principle: utilitarian, equitable) × 4 (Group: race, disability, elderly, addiction) experimental design. Participants read a scenario and answered questions similar to those used in Experiments 1 and 2. In each scenario, a hospital adopts a triage policy that allocates scarce lifesaving resources to those with the best chances of recovery, consistent with the principle of maximizing benefits. The triage team then discovers that this policy disadvantages members of certain at-risk or historically marginalized groups. Specifically, they discover that because members of these groups have a lower chance of recovery than others, they end up receiving less priority for lifesaving resources. Participants were randomly assigned to conditions varying the group that was affected (e.g. people of color, people with disabilities, the elderly, or people with substance use disorders). Participants were also randomly assigned to conditions manipulating the way that the triage team reacts to the discovery that allocation disadvantages these groups. In one condition, the team reaffirms the utilitarian principle that perpetuates the disadvantaged outcome. In the other condition, the team adopts a policy that gives the same priority as others to disadvantaged groups. Below are examples taken from the race utilitarian condition and the elderly equitable condition:

**Race utilitarian condition.** The triage team discovers that due to prior risk factors, people of color have a lower chance of recovering from the infection and are thus not receiving the same priority as others for lifesaving resources. The triage team commissions a study into the causes of prior risk factors. But they do not change their policy so that people of color receive the same priority as others. As a result of this, fewer people of color will recover from the infection but more people will recover in total.

**Elderly equitable condition.** The triage team discovers that due to prior risk factors, elderly people have a lower chance of recovering from the infection and are thus not receiving the same priority as others for lifesaving resources. The triage team commissions a study into the causes of prior risk factors. And they also change their policy so that elderly people receive the same priority as others. As a result of this, more elderly people will recover from the infection but fewer people will recover in total.

Beneath the scenario, on the same page, participants responded to the following items:

1. The triage team changed their policy regarding lifesaving resources. (comprehension 1)

2. The policy the triage team decided on is a good one. (test 1)

3. I would choose this hospital over others if I or someone I loved needed lifesaving treatment. (test 2)

Responses to the comprehension check were collected by a dichotomous yes/no answer. Items two and three were evaluated with Likert scales identical to those used in Experiments 1 and 2.

## Results

Our research question asked how participants would evaluate a utilitarian allocation principle when it disadvantaged at-risk or historically marginalized patient groups. To answer this question, we first excluded those participants who did not pass the comprehension check (see pre-registration). One hundred and fifteen participants were removed for not answering this correctly, leaving a total sample of five hundred and twenty-four participants. We examined the relationship between responses to our two dependent variables, policy and hospital, using a Spearman-Brown prediction formula. The Spearman-Brown test coefficient was .885. We averaged these variables into a single combined measure for analysis. Multiple linear

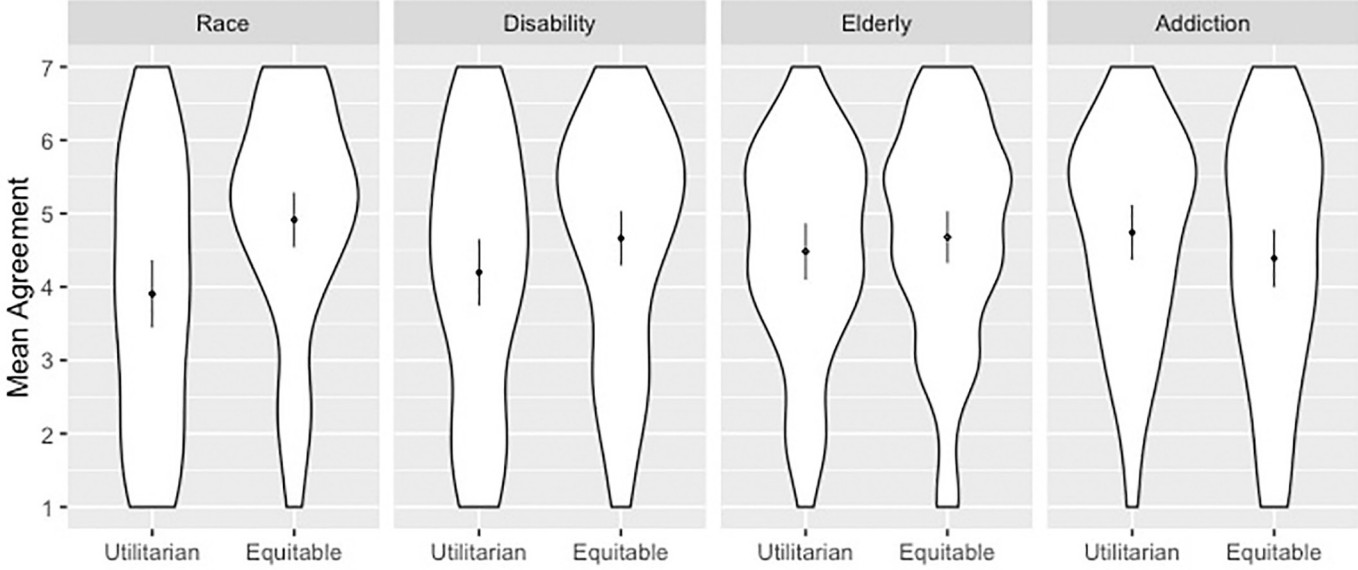

**Fig 3. Mean response for each group grouped by principle.** Scales ran 1 (SD)—7 (SA). Dots overlay distributions and show means with 2 SEM.

regression was conducted with the combined measure as the dependent variable and with allocation principle, patient group, an interaction of principle and group, self-reported biological sex, and age as predictors.

There was a significant effect for allocation principle, patient group, and their interaction (Fig 3 and Table 9). Principle was a significant predictor of responses, with greater scores when the hospital switched allocation policy as compared to upholding utilitarian allocation and allowing bias to continue ($\beta = 1.02$). Patient groups also predicted responses with higher scores associated with elderly ($\beta = .60$) and addiction ($\beta = .86$) conditions compared to the race condition. The differential impact of principle on the elderly ($\beta = -.82$) and addiction ($\beta = -1.39$) conditions was associated with lower responses as compared to the race condition.

Follow-up independent samples t-tests revealed that the principle/group interaction occurred because adopting a triage policy that gave the same priority as others to

**Table 9. Experiment 3: Multiple linear regression predicting the combined measure.**

| Predictors | Beta | CI | p |
|---|---|---|---|
| (Intercept) | 3.62 | 2.97 4.28 | < .001 |
| Principle | 1.02 | 0.47 1.57 | < .001 |
| Group | | | |
| Disability | 0.31 | -0.26 0.88 | .285 |
| Elderly | 0.60 | 0.01 1.18 | .045 |
| Addiction | 0.86 | 0.29 1.44 | .003 |
| Sex | < 0.01 | -0.29 0.28 | .997 |
| Age | 0.01 | -0.00 0.02 | .230 |
| Principle*Disability | -0.57 | -1.35 0.21 | .151 |
| Principle*Elderly | -0.82 | -1.61–0.03 | .041 |
| Principle*Addiction | -1.39 | -2.17–0.60 | .001 |

N = 524, $R^2$ = 0.038, Adjusted $R^2$ = 0.021. Reference class for Principle is Utilitarian; reference class for Group is Race; reference class for Sex is Female.

**Table 10. Experiment 3: Independent sample t-tests.**

|  | Utilitarian | | Equitable | | | | | | | |
|---|---|---|---|---|---|---|---|---|---|---|
|  | M | SD | M | SD | t | df | p | MD | 95% CI | d |
| **Race** | 3.90 | 1.81 | 4.91 | 1.56 | -3.429 | 129 | .001 | -1.01 | -1.59–0.43 | -0.60 |
| **Disability** | 4.20 | 1.77 | 4.66 | 1.60 | -1.605 | 133 | .111 | -0.47 | -1.04 0.11 | -0.28 |
| **Elderly** | 4.48 | 1.43 | 4.68 | 1.51 | -0.744 | 126 | .458 | -0.20 | -0.72 0.33 | -0.13 |
| **Addiction** | 4.74 | 1.43 | 4.39 | 1.67 | 1.275 | 128 | .205 | 0.35 | -0.19 0.90 | 0.22 |

**Table 11. Experiment 3: One sample t-tests.**

|  | t | df | p | MD | 95% CI | d |
|---|---|---|---|---|---|---|
| **Utilitarian** |  |  |  |  |  |  |
| Race | -0.42 | 61 | .676 | -0.10 | -0.56 0.36 | 0.05 |
| Disability | 0.87 | 60 | .389 | 0.20 | -0.26 0.65 | -0.11 |
| Elderly | 2.50 | 54 | .016 | 0.48 | 0.10 0.87 | -0.34 |
| Addiction | 3.95 | 57 | < .001 | 0.74 | 0.37 1.12 | -0.52 |
| **Equitable** |  |  |  |  |  |  |
| Race | 4.87 | 68 | < .001 | 0.91 | 0.54 1.29 | -0.59 |
| Disability | 3.57 | 73 | .001 | 0.66 | 0.29 1.03 | -0.41 |
| Elderly | 3.84 | 72 | < .001 | 0.68 | 0.33 1.03 | -0.45 |
| Addiction | 1.98 | 71 | .052 | 0.39 | -0.003 0.78 | -0.23 |

disadvantaged groups did not significantly impact mean responses between disability, elderly, or addiction conditions, but it did impact responses between race conditions (t(129) = -3.429, p = .001, d = .06) (Table 10).

Follow-up one sample t-test showed that when a hospital adhered to utilitarian allocation, responses were significantly above the neutral midpoint of 4 in elderly and addiction conditions (Table 11). When the hospital revised priority allocation, however, responses were significantly above the midpoint for all conditions except for addiction.

An exploratory analysis was conducted to determine the influence of demographic and personal factors on responses. We conducted a multiple regression analysis with the combined measure as the dependent variable that included principle, group, an interaction of principle and group, race, sex, age, highest level of education completed, religiosity, social liberalism, economic liberalism, hospital experience, and hospital training as predictors. Religiosity predicted responses, where higher response ("I consider myself a religious person") was associated with higher scores on the combined measure ($\beta$ = .14, t (506) = 4.36, p < .001). Hospital experience (e.g. "Have you been admitted to the hospital any time in the last 5 years?") was also associated with higher scores ($\beta$ = .33, t (506) = 2.18, p = .03). There was weak evidence for an effect of general level of education, with higher levels of education associated with higher scores ($\beta$ = .11, t (506) = 1.72, p = .0857). No effects were observed with other demographic or personal factors. These effects are exploratory, have not been adjusted for multiple comparisons, and are only included here to guide future research.

## Discussion

This experiment examined how participants evaluated utilitarian resource allocation when it disadvantaged at-risk or historically marginalized patient groups. We found that participants' attitudes depended on the patient group affected. Participants significantly approved of

utilitarian allocation even if it disadvantaged the elderly and those with substance use disorders. When allocation disadvantaged people of color or people with disabilities, participants were split, neither significantly agreeing nor disagreeing. However, when utilitarian allocation that disadvantaged people of color was revised in favor of allocation that gave them the same priority as others, this received significantly more agreement than when the policy was not revised (d = 0.60). Lastly, we found that participants negatively evaluated equitable resource allocation that advantaged patients with substance use disorders, though this trend was not statistically significant.

## General discussion

This study investigated public attitudes of U.S. residents toward ethical principles for allocating scarce resources in the COVID-19 pandemic. Participants generally agreed with policies motivated by principles favoring resource allocation to patients with the best chances of surviving treatment when resources were scarce, but they also agreed with allocation that prioritized patients who were the worst-off. Although participants agreed with these principles during initial allocation, they were ambivalent about them when considering reallocation of scarce resources between existing patients. In these conditions, participants agreed with a reallocation policy that prioritized patients who received the resources first during initial allocation. Across experiments, participants strongly disagreed with allocation based on social usefulness. Lastly, participants were ambivalent toward utilitarian allocation if people of color and people with disabilities were disadvantaged. However, participants approved of utilitarian allocation, even if it disadvantaged the elderly or people with substance use disorders. We also found that revising a utilitarian allocation policy that disadvantaged people of color by replacing it with one that assigned them the same priority as others resulted in significantly higher agreement with that policy.

These findings provide preliminary insight into public attitudes toward the ethical principles of resource allocation in the COVID-19 pandemic. Utilitarian allocation, or the maximizing benefits principle, serves as a backbone for many—if not most—current triage policies. Although participant attitudes were complex and multifaceted, we found less support for utilitarian allocation than in previous studies [21, 27]. For example, using survey designs featuring non-clinical "life-boat dilemmas" [28], researchers have found that laypeople favor saving many lives over saving one drowning victim [27]. These researchers also found similar results using clinical scenarios. When allocating scarce resources between two critically-ill newborns, most participants choose to allocate to the infant with the best chances of survival. These findings are also partially consistent with prior research on public attitudes toward resource allocation in public health emergencies. Daugherty Biddison and colleagues conducted a series of stakeholder engagement activities designed to cultivate deliberative democracy on triage policies [21]. They found that, in Howard County, Maryland, 83% of participants said utilitarian allocation is "always or often" acceptable, whereas only 33% of participants from East Baltimore, an area with many traditionally Black communities, agreed.

Researchers have also found support for utilitarian allocation among health care professionals [22]. Utilizing a modified Delphi procedure [see 29], Christian and colleagues found that experts agreed that patients should be excluded from receipt of scarce resources in public health emergencies if they "have such a low probability of survival that significant benefit is unlikely" [22]. This coheres with several other expert recommendation documents published by national medical organizations [7, 30–32].

There are several reasons why we may have detected lower approval or ambivalence concerning utilitarian allocation relative to previous studies. First, the present study was

conducted in the midst of the COVID-19 pandemic, one of the worst and enduring public health emergencies in the past 100 years. Previous survey research was conducted during outbreaks of MERS-Coronavirus, Ebola, and Avian Flu, but these emergencies lacked the pointed and concrete threat to resource allocation as presented by COVID-19.

Second, weakest support for utilitarian allocation was found in cases involving reallocation and disadvantage, which were not the focus of prior studies [although see 23 for concern over withdrawing ventilators]. Indeed, these issues have been at the forefront of media, scholarly, and government reports in the current pandemic. In the United States, people of color have increased COVID-19 mortality due to prevalent comorbidities, such as diabetes or hypertension, which often result from unjust structural inequalities in social determinants of health [33]. Moral intuitions to mitigate or redress this injustice may underly anti-utilitarian attitudes.

Third, the present results might also vary based on background and demography of participants in ways that we did not detect. In contrast, some previous research has examined attitudes germane to geographically, racially, and socioeconomically defined communities, finding significant differences in attitudes between groups. In at least one study, participants from White socioeconomically advantaged communities strongly favored utilitarian principles, whereas participants from historically marginalized communities did not [21].

Fourth, different results may obtain when broadening allocation cases to consider conflicting values and principles. Subsequent research by Huang and colleagues, examining allocation of scarce ventilators during the COVID-19 pandemic, also found mixed support for utilitarian allocation [34]. When forced to choose between a 25-year-old patient or a 65-year-old patient who arrived at the hospital first, participants judged that it was morally acceptable for doctors to allocate ventilators to the younger patient over the older patient at chance rates. However, researchers also found that participants judged allocation to the younger patient more morally acceptable when they were first encouraged to engage in veil of ignorance reasoning. This study is notable as it suggests at least some initial uncertainty or resistance to utilitarian allocation that maximises the number of life-years saved, but that these attitudes might change after small interventions.

We also found evidence that support for allocation principles is affected by allocation stage. Although participants agreed with prioritizing the worst-off and utilitarian principles during initial allocation, egalitarianism was the only principle that was viewed more favorably in subsequent allocation stages—namely, in the dilemma of forcibly withdrawing scarce resources from one patient to give to another. This effect may be due to intuitions that the forcible withdrawal of lifesaving resources is tantamount to killing. Experts argue that, in a pandemic, forcible withdrawal is justified because it is not theoretically different from withholding resources [12, 35]. But even if this is true in theory, the public may disagree in practice [36]. Indeed, recent work on clinician and layperson attitudes toward notions of "killing" and "letting die" demonstrates that the distinction rests on foundational moral and causal judgments in social cognition [37]. Specifically, researchers found that, if a patient with a terminal diagnosis wants to die during the withdrawal of life-prolonging treatment, the cause of death is associated with the patient's disease. If the patient does not want to die, the cause of death is associated with clinicians' behavior. To the extent that these processes bear on reallocation judgments, this might explain resistance to forced reallocation. At the same time, however, it is important not to exaggerate this resistance. Given the controversial nature of withdrawing treatment due to resource scarcity, the fact that participants appeared evenly split in their evaluations might also suggest more support for reallocation than would otherwise be expected.

Our findings also suggest that support for utilitarian allocation varies when it disadvantages some groups over others. Approval of utilitarian allocation that disadvantages the elderly may

be due to intuitions about the number of "life-cycles" a patient has experienced. Studies by Daugherty Biddison and colleagues [21] and Huang and colleagues [34] show that laypeople will allocate resources to younger patients in certain conditions, potentially reflecting the belief that the quantity of life-years saved is just as important as the quantity of lives saved. Additionally, approval of utilitarian allocation that disadvantages people with substance use disorders might be due to latent negative stigmas toward drug use and addiction. One possibility is that laypeople perceive this behavior as a choice, whereas disability and racial disparities in the social determinants of health are not. This may lead laypeople to tolerate some disadvantages generated by utilitarian allocation over others.

Lastly, these findings may improve our understanding of the ethical principles used to investigate, construct, and debate triage policies. On a theoretical level, there appears to be significant overlap between some principles when justifying triage policies. For example, resource allocation based on social usefulness can align with the principle to maximize benefits in the sense that prioritizing health care workers during a pandemic could ultimately maximize the number of lives saved. Similarly, in some cases, triage policies that prioritize patients who are the worst-off might ultimately be the best way to save the greatest number of lives or it might correlate with age or other socially significant demographic categories. While alignment among disparate allocation principles is indeed possible in theory and practice, this may not be reflected in ordinary psychological judgments. To the extent that participants drew distinctions between approaches to resource allocation, it suggests that they may evaluate and process allocation principles differently.

## Limitations

This study has several limitations. First, participants were recruited using Mechanical Turk, which has strengths and weakness relative to non-web-based sampling and should be replicated in other samples before drawing strong conclusions [38, 39]. Furthermore, the study was restricted to participants residing in the United States and may not generalize to other countries or infection hotspots where health care resources are scarce.

Second, while participants were recruited from a national sample, this may obscure the needs of individual communities. COVID-19 will stress communities and health systems differently throughout the pandemic. It is a matter of debate whether triage policies ought to be standardized across communities and health systems, as drawing inferences from a national sample might suggest.

Third, while we investigated several principles of resource allocation, we only focused on a small number of interpretations with respect to specific policies. For example, a different way to assess support for policies motivated by the principle of maximizing benefits might be to specify benefits in terms of the number of life-years saved rather than total lives. Similarly, there are many other ways researchers might specify who is "worst-off" when assessing support for this principle, such as susceptibility to illness or other prior risk factors.

Relatedly, there has been significant discussion in philosophy and cognitive science surrounding the labeling and dimensions of ethical categories [8, 40,41]. On a theoretical level, some researchers question whether the maximizing benefits principle should be classified as "utilitarian", when non-utilitarian moral considerations can equally support the goal of saving the most lives or life-years [40]. On an empirical level, work in moral psychology has also questioned the extent to which judgments in sacrificial moral dilemmas to save the most lives actually reflect utilitarian decision-making and the desire to maximize aggregate welfare [42]. These concerns highlight an opportunity for future research on variations in labeling of ethical

categories, how they influence judgments of triage policies, as well as the underlying psychological processes that generate them.

Fourth, our study did not explore evaluations of triage policies explicitly informed by multiple principles. Optimal triage policies will no doubt involve a multi-principle approach [9]. For example, initial allocation of resources might follow from the maximizing benefits principle, but granular allocation decisions between patients, or "tie breakers", might follow from social usefulness or egalitarian principles. The design of the present study only allowed participants to express views about a single approach, which may obscure relative preferences between allocation principles or their interaction. Future work that assesses the complexity of complete triage policies, rather than distinct underlying principles, might reveal these relationships.

## Conclusion

This study furthers our understanding of attitudes toward ethical principles underlying triage policies during the COVID-19 pandemic. Understanding these attitudes can provide valuable insight into how health systems can better serve the public. We do not necessarily recommend that health policies be adopted solely based on public opinion. However, policies and practices that depart radically from public opinion are unlikely to be regarded as optimal or fully ethically legitimate. Triage policies can be enhanced by consulting the communities impacted by them. Engaging communities helps policymakers identify competing values, such as the duties to treat all patients fairly versus the stewardship of scarce resources. Moreover, no approach to triage will function well if the public is discouraged from presenting at hospitals that adopt negatively viewed policies. The ability to anticipate potential resistance to triage policies can assist health systems improve public relations and communicate their rationale more effectively.

## Supporting information

**S1 File. Sample demographics.**
(PDF)

**S2 File. Survey materials.**
(PDF)

## Acknowledgments

For helpful and incisive feedback, we thank Carolyn Buckwalter, Zachary Horne, Jason Karlawish, Josh Knobe, Jonathan Lewis, Jonathan Livengood, Govind Persad, John Turri, Charles Weijer, and Dominic Wilkinson.

## Author Contributions

**Conceptualization:** Wesley Buckwalter, Andrew Peterson.

**Data curation:** Wesley Buckwalter.

**Formal analysis:** Wesley Buckwalter.

**Funding acquisition:** Andrew Peterson.

**Investigation:** Wesley Buckwalter.

**Methodology:** Wesley Buckwalter, Andrew Peterson.

**Project administration:** Andrew Peterson.

**Visualization:** Wesley Buckwalter.

**Writing – original draft:** Wesley Buckwalter, Andrew Peterson.

**Writing – review & editing:** Wesley Buckwalter, Andrew Peterson.

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
