## [Decision Letter · Decision Letter 0]

17 Sep 2020

PONE-D-20-19914

Public Attitudes Toward Allocating Scarce Resources in the COVID-19 Pandemic

PLOS ONE

Dear Dr. Peterson,

Thank you for submitting your manuscript to PLOS ONE. After careful consideration, we feel that it has merit but does not fully meet PLOS ONE’s publication criteria as it currently stands. Therefore, we invite you to submit a revised version of the manuscript that addresses the points raised during the review process.

You will see below that Reviewer 2 asks for some minor revisions.

We look forward to receiving your revised manuscript.

Kind regards,

Noam Lupu

Academic Editor

PLOS ONE

Journal Requirements:

2. Please amend either the title on the online submission form (via Edit Submission) or the title in the manuscript so that they are identical.

Reviewers' comments:

Reviewer's Responses to Questions

**Comments to the Author**

1. Is the manuscript technically sound, and do the data support the conclusions?

Reviewer #1: Yes

Reviewer #2: Yes

2. Has the statistical analysis been performed appropriately and rigorously? 

Reviewer #1: Yes

Reviewer #2: I Don't Know

3. Have the authors made all data underlying the findings in their manuscript fully available?

Reviewer #1: Yes

Reviewer #2: Yes

4. Is the manuscript presented in an intelligible fashion and written in standard English?

Reviewer #1: Yes

Reviewer #2: Yes

5. Review Comments to the Author

Reviewer #1: This extremely well-presented paper is vitally important for several reasons. Firstly, it displays an excellent level of technical and manipulative soundness in terms of the material preparation, data collection and data analysis. This reviewer was particularly impressed by the transparency of, and detail in, the presentation of the data handling and analysis elements. Secondly, in this this reviewer's opinion, it is a excellent example of experimental philosophical bioethics ('bioxphi'), a burgeoning field of study derived from the methods and approaches in experimental philosophy ('x-phi') and experimental psychology. Indeed, this reviewer hopes that it will be received as a paradigmatic case of quality bioxphi by other researchers working in this up-and-coming field. Thirdly, and perhaps most importantly, even though there is already a massive literature on the ethics of scarce ICU resources and the triaging of COVID-19 patients, this paper is particularly novel because it demonstrates the ways in which the public respond to the disparate and (on the whole) poorly justified and articulated ethical values to which health organisations have merely paid lip service in the development of their respective COVID-19 clinical guidelines. Moreover, it goes some way to fulfilling the calls made by medical ethicists, health policy researchers and legal scholars for greater public input into the formation of COVID-19-related clinical guidelines, particularly as they pertain to policy and procedures regarding triage admission/exclusion and resource distribution. Furthermore, the particularly nuanced approach to understanding public attitudes to the various ethical principles under different triage conditions calls into question the totalising utilitarian and egalitarian approach advocated by internationally-recognised health policy and medical ethics experts. Thus, the findings could offer policymakers and health practitioners greater clarity of how different triage and allocation protocols are needed in different situations in order to respond to the competing ethical demands of extant clinical guidance. Overall, it was an absolute pleasure to read such a fine, well-crafted, well-explained and well-discussed piece of research, which has genuine implications not only for how medical ethicists and health policy researchers might (and perhaps should) think about and respond to the ethical dimensions of pandemic emergency situations, but for how future clinical guidelines and health policy should be developed in a way that responds to public attitudes.

Reviewer #2: PLOS One review

This paper investigates US residents’ attitudes toward ethical principles underlying triage policies as related to the COVID-19 pandemic. It makes a timely and important contribution that addresses a gap in knowledge, and potentially contributes to a crucial and ongoing public debate, especially given the possibility of second waves or ongoing surges of the virus, or any future pandemic. Its publication would be a valuable addition to the literature on public attitudes toward triage policies, though there are significant points that require revision and clarification first.

Major comments

1. Accuracy and interpretation of “prioritarian policy”. One of the four triage policies assessed in the paper is labelled “prioritarian” – directing treatment preferentially to those most seriously ill. Participants were told that “patients will receive lifesaving resources in the order of the seriousness of illness, with those who are the worst off being prioritized.” Throughout the abstract/results/discussion approval of this policy is interpreted as reflecting views about prioritarian triage. However, there is a problem in that the policy as described is not straightforwardly prioritarian. Priority based on “clinical need” or severity of illness is a common existing approach to triage – however, it is potentially justified on utilitarian not purely prioritarian grounds. (Eg Those patients who are most severely ill are least likely to be able to cope without urgent attention in the emergency department or on the battlefield). If a group of patients are all going to die without respiratory support, it is not clear that those who are “most seriously ill” are worst off – from a prioritarian perspective. In the introduction, the authors refer to priority to those who are “medically most vulnerable”. That might include giving priority to those with most co-morbidity, disability, frailty etc prior to contracting COVID. That would be a prioritarian approach, but it is not what survey respondents were asked to approve/disapprove of. Suggestion: Avoid equating triage based on seriousness of illness with “prioritarianism”. Refer in abstract and discussion to priority based on seriousness/severity of illness. Discuss in general discussion and limitations that triage policies were a simplification and do not necessarily match what ethical theory would dictate.

Supplementary suggestion: There is a related (though less blatant) problem in the description of “utilitarian” triage policy. A utilitarian approach to allocation of ventilators would not necessarily or simply favour saving the most lives. (It would also consider the length of lives saved and their quality of life) See for example https://onlinelibrary.wiley.com/doi/10.1111/bioe.12771 (not suggesting that you need to cite this paper, but just to illustrate)

2. Care over interpretation of findings and language. In the abstract and discussion, the authors indicate that survey respondents “did not agree” with utilitarian policy that disadvantaged at risk groups, and they imply a lack of support for utilitarian [actually maximising survival] reallocation of treatment. However, the most accurate way of describing the views about reallocation of treatment is that participants were evenly divided in their views about reallocation of treatment in order to save the most lives. From figure 2/table 7 it seems that the range of agreement was right across the spectrum with similar numbers of respondents agreeing and disagreeing (mean response 3.86 – close to the mid point). The authors have interpreted this negatively. In the discussion they state (unpersuasively) that their results “suggests that there will be deep resistance to forced reallocation”. (Elsewhere they refer to “the strongest opposition to utilitarian reallocation”). But although opposition/resistance is possible, it just isn’t clear that their results show that. Indeed, given the controversial nature of decisions about withdrawing treatment for reasons of limited resources, it is interesting just how evenly split views are on the topic. That might suggest more support for reallocation than would be expected. [Also, perhaps clarify? the authors seem to be making much of a small shift in the mean approval on independent t-test, but the multiple linear regression in table 6 implies that reallocation for the lives principle was not significantly associated with a changed approval?]

Suggestion: Review and revise the language of the abstract and discussion in relation to interpretation of lack of support for policies – eg “utilitarian” reallocation – this also applies to discussion of policies that disadvantage specific groups. One example of a better description in the paper is on page 26 “participants were ambivalent towards util allocation” – I think that sort of language applies to a number of other findings relating to disadvantaged groups.

Supplementary suggestion: In several places, the authors refer to “abandoning utilitarian allocation” leading to increased agreement. I have two concerns about this phrase. First – methodological: in this survey participants were randomised and only expressed views about one policy variation. Therefore, I think it is not accurate to say that a particular change in policy led to higher endorsement. It would be more accurate to say that respondents who viewed policies that had shifted away from ‘life-maximising’ triage (where that disadvantaged patients of color) expressed higher levels of agreement with the policy. [Parenthetically, I think the authors should consider discussing as a limitation that since individual participants only expressed views about a single policy, it is not possible to make direct inferences about relative preferences – ie it is not possible to say whether participants “prefer” one policy over another – only that those who saw a particular policy evinced higher or lower support than others who saw a different policy] Second – terminological. I think that the language of “abandoning utilitarian allocation” might be loaded. It would be more accurate/neutral to note that policies that were revised in favour of equal allocation received higher ratings of approval or something similar…

3. Attending hospitals. Participants were asked whether they agreed or disagreed with a particular policy and also whether they would attend a hospital with a particular policy. It wasn’t totally clear to me why the authors merged these two answers in their analysis. I suspect it is because answers were virtually identical. However, I think it would be worth clearly explaining that. [Parenthetically, in the very end of the paper, the authors refer to patients “categorically refusing to attend hospitals”. While this is rhetorically powerful, it isn’t clear that any of the policies in the paper (except priority to “important people”) would lead to large numbers of patients refusing to attend – but it is hard to comment on this since the separate results for hospital preference are not given.]

4. US specific interpretation. As a limitation, it would be important to acknowledge that these surveys were conducted with US citizens and may not be applicable to other countries. (Perhaps particularly given the political and healthcare climate in the US). It might be worth making a note about the timing of the survey in relation to race/riots, and George Floyd – that is obviously relevant to some of the questions relating to race and discrimination.

Minor comment

- A forthcoming study by Huang and colleagues is described (bottom of p.28) as finding less support for utilitarian allocation, noting that participants were only inclined to allocate scarce lifesaving resources to young patients after an intervention that primes utilitarian thinking. It is not entirely clear what the priming intervention refers to, but presumably it is the veil of ignorance condition deployed in that study. This seems misleading as the veil of ignorance is deployed to attenuate self-serving bias, not prime utilitarian thinking.

Suggestion: Revise the phrasing to reflect that participants more strongly favoured a policy that maximised the number of life-years saved by being primed to think in a less self-serving manner via a veil of ignorance condition.

6. PLOS authors have the option to publish the peer review history of their article (what does this mean?). If published, this will include your full peer review and any attached files.

Reviewer #1: **Yes: **Jonathan Lewis

Reviewer #2: **Yes: **Dominic Wilkinson

---

## [Author Response · Author response to Decision Letter 0]

26 Sep 2020

We are grateful for the opportunity to submit a revision of this manuscript to PLOS ONE. We are pleased that the reviewers’ assessed the manuscript as an “extremely well-presented paper” that represents a “timely and important contribution” to the literature. We also appreciate that Reviewer 2 not only raised several insightful criticisms, but also offered concrete suggestions to address them. In the following report, we review the main points raised by Reviewer 2 and explain how we have responded to them. We have instituted the suggested changes in all cases, and, in our judgment, this has significantly improved the manuscript.

First, Reviewer 2 writes that the wording used to describe the lack of support for policies in the abstract and discussion could be misleading in cases where participants are ambivalent and, relatedly, that participant resistance to utilitarian reallocation is overstated. We agree with Reviewer 2 and have made revisions throughout the manuscript to avoid this. In the abstract (p. 2) and experimental discussion sections (pp. 13, 20, 26) we have revised the language regarding interpretations of participant ambivalence. We also call attention to the fact that resistance to utilitarian reallocation is not as strong as one might have predicted given the controversial nature of withdrawing life-sustaining therapy (p. 30). 

Related to this point, the reviewer also notes that the phrase “abandoning utilitarian allocation” could be misleading because participants only expressed views about one variation in allocation principles. Reviewer 2 suggests that this be noted as a limitation. In the revised manuscript, we remove this phrase and note more explicitly this fact as a limitation (p. 32). 

Second, Reviewer 2 objects to our use of the term “prioritarian” to describe a policy that prioritizes patients who are the worst-off in terms of the severity of illnesses. Reviewer 2 reasons that: (i) there are other ways that patients might be considered worst-off, and (ii) participants might have ultimately been thinking about allocation on utilitarian grounds rather than prioritarian grounds. Review 2 also makes a similar, but minor, point regarding our presentation of the maximizing benefits principle, where maximizing benefits could be interpreted as saving the most lives, the most life years, or a combination of both. 

We agree that participants may have had various interpretations of these ethical principles and that these interpretations may not reflect the way some theorists classify prioritarian or utilitarian theories. To address these concerns, we make several revisions to the manuscript: 

• We remove the term “prioritarianism” from the manuscript as Reviewer 2 suggests. We replace this term with variations of “worst-off” or “seriousness of illness” that better align with the language used in our experimental vignettes. 

• We re-write the paragraph(s) in the introduction introducing ethical principles (p. 3). This now makes it clear that (i) ethical principles are used to motivate specific triage policies, (ii) triage policies motivated by these principles might be instituted in a number of different ways, and (iii) there is scholarly debate about which triage policies best capture underlying allocation principles in practice. 

• We expand the limitations section to explicitly discuss disagreements about ethical principles, using utilitarianism as an example (p. 32). We note that there is an important theoretical disagreement concerning which policy should rightfully be called “utilitarian” as well as discuss important experimental reasons to question whether underlying psychological processes associated with “utilitarian” answers truly reflect utilitarian reasoning. 

• We add a paragraph to the general discussion exploring the hypothesis that participants interpreted allocation based on severity of illness as justified by maximizing benefits (p. 31). Though this could be a natural interpretation, interestingly, to the extent that participants evaluate these policies differently, there is some evidence to suggest that they were not processing the cases in that way.

Third, Reviewer 2 writes that “As a limitation, it would be important to acknowledge that these surveys were conducted with US citizens and may not be applicable to other countries”. We have added this to the list of study limitations on p. 31.

Fourth, Reviewer 2 writes that the description of prior work by Huang and colleagues may be misleading because veil of ignorance manipulations are not usually taken to prime utilitarian thinking. We correct this description on p. 29. 

Fifth, Reviewer 2 asks why dependent variables were combined for analysis. In the revised manuscript we clarify that this combination was specified in our preregistration document to a create a scale if responses to different conditions were extremely similar because items were tapping into the same underlying construct. However, the data are freely available through the Open Science Framework if others wish to explore the data along these dimensions separately. Along with this, we also remove the phrase “categorically refusing to attend hospitals” singled out by the reviewer.

---

## [Editor Report · Decision Letter 1]

1 Oct 2020

Public Attitudes Toward Allocating Scarce Resources in the COVID-19 Pandemic

PONE-D-20-19914R1

Dear Dr. Peterson,

We’re pleased to inform you that your manuscript has been judged scientifically suitable for publication and will be formally accepted for publication once it meets all outstanding technical requirements.

Kind regards,

Noam Lupu

Academic Editor

PLOS ONE
---

## [Editor Report · Acceptance letter]

28 Oct 2020

PONE-D-20-19914R1 

Public attitudes toward allocating scarce resources in the COVID-19 pandemic 

Dear Dr. Peterson:

I'm pleased to inform you that your manuscript has been deemed suitable for publication in PLOS ONE. Congratulations! Your manuscript is now with our production department. 

Kind regards, 

on behalf of

Dr. Noam Lupu 

Academic Editor

PLOS ONE